# IRSDD-YOLOv5: Focusing on the Infrared Detection of Small Drones

**Shudong Yuan, Bei Sun \*, Zhen Zuo, Honghe Huang, Peng Wu, Can Li, Zhaoyang Dang and Zongqing Zhao**

College of Intelligence Science and Technology, National University of Defense Technology, Changsha 410073, China; yuanshudong21@nudt.edu.cn (S.Y.); z.zuo@nudt.edu.cn (Z.Z.); huanghonghe16@nudt.edu.cn (H.H.); pengwu@nudt.edu.cn (P.W.); lican22@nudt.edu.cn (C.L.); dzy0329@nudt.edu.cn (Z.D.); zhaozongqing17@nudt.edu.cn (Z.Z.)
* Correspondence: sunbei08@nudt.edu.cn

**Abstract:** With the rapid growth of the global drone market, a variety of small drones have posed a certain threat to public safety. Therefore, we need to detect small drones in a timely manner so as to take effective countermeasures. At present, the method based on deep learning has made a great breakthrough in the field of target detection, but it is not good at detecting small drones. In order to solve the above problems, we proposed the IRSDD-YOLOv5 model, which is based on the current advanced detector YOLOv5. Firstly, in the feature extraction stage, we designed an infrared small target detection module (IRSTDM) suitable for the infrared recognition of small drones, which extracted and retained the target details to allow IRSDD-YOLOv5 to effectively detect small targets. Secondly, in the target prediction stage, we used the small target prediction head (PH) to complete the prediction of the prior information output via the infrared small target detection module (IRSTDM). We optimized the loss function by calculating the distance between the true box and the predicted box to improve the detection performance of the algorithm. In addition, we constructed a single-frame infrared drone detection dataset (SIDD), annotated at pixel level, and published an SIDD dataset publicly. According to some real scenes of drone invasion, we divided four scenes in the dataset: the city, sky, mountain and sea. We used mainstream instance segmentation algorithms (Blendmask, BoxInst, etc.) to train and evaluate the performances of the four parts of the dataset, respectively. The experimental results show that the proposed algorithm demonstrates good performance. The $AP_{50}$ measurements of IRSDD-YOLOv5 in the mountain scene and ocean scene reached peak values of 79.8% and 93.4%, respectively, which are increases of 3.8% and 4% compared with YOLOv5. We also made a theoretical analysis of the detection accuracy of different scenarios in the dataset.

**Keywords:** drone defense; small target detection layer; instance segmentation; dataset



## 1. Introduction

Drone detection in complex, low-altitude environments is an important research area. Visible light images have high resolutions and can capture drone target detail information better, but visible-light-based drone detection systems cannot work in night scenes and low-light conditions. Millimeter-wave radar has a long detection range and wide coverage, but small quadrotor drones have a small reflection cross-section for millimeter-wave radar, making millimeter-wave-radar-based drone detection systems unable to detect drones better. Additionally, millimeter-wave radar is expensive. Infrared sensors are very sensitive to the detection of heat sources and can measure the temperatures of objects at a certain distance without making contact with the target. A working drone acts as a source of heat radiation and will continuously radiate heat outward in flight, so infrared-thermography-based detection is an important sensing method for defense against low-altitude drones, especially in dark environments.

The accurate detection and identification of drones is extremely challenging due to three main factors. The first factor is that the air contains infrared interference sources

such as high, bright backgrounds with radiation sources, the background edges of complex clouds and blind flash sourcein the system. The second factor is the small size of the drone target and the fact that one always wants to find the target at a longer distance in order to confirm the intrusion of the target as early as possible; such targets are imaged in the image with a smaller area after being imaged by the infrared system. The third factor is the poor quality of the resulting infrared image due to the sensitivity of the detection device, atmospheric scattering, background temperature noise and other factors. We show four infrared small drone targets in a complex scene at a low altitude in Figure 1. The first image contains a ship in a complex urban scene with buildings in the background; the second image presents a mountainous scene with numerous miscellaneous sources of radiation similar in shape to the drone; the third image contains a faint, five-pixel-sized target in a sea scene; the fourth image contains a highlighted target against a cloudy background. The infrared images of the targets to be detected contain weak signals and lack contour and texture information.

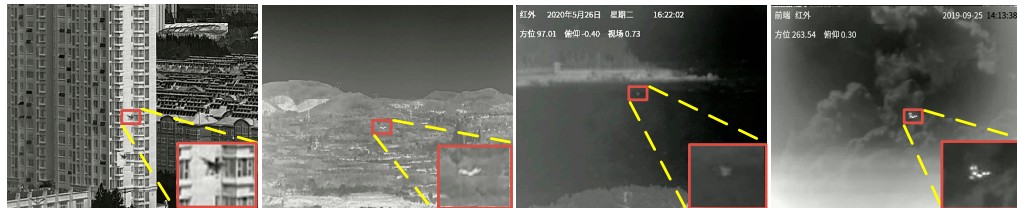

**Figure 1.** Example image of the infrared small drone. The drone is represented by a red border and enlarged in the lower right corner.

There are a number of traditional infrared target detection methods that can be used to detect drones, including filter-based [1], patch-image [2], and rank-based [3] methods. The detection performances of traditional detection methods rely heavily on the effectiveness of manually extracting target features, which is insufficient for use in complex scenarios. In particular, for drones flying at low altitudes with large target background differences and unpredictable scene information, traditional detection methods cannot guarantee that they will maintain their robustness in extreme environments. Therefore, the traditional methods based on the manual extraction of target features are not suitable for practical applications.

The current target detection methods based on convolutional neural networks (CNNs) have great potential because convolutional neural networks demonstrate strong representability and can effectively extract target features for learning and training detection algorithms [4–7]. The convolutional neural network algorithm effectively combines the candidate box and a convolutional neural network, making a significant breakthrough in the field of target detection. In the task of drone defense and identification based on infrared images, due to the existence of multiple scenes and multiple drone spans, which cause the target to present in a weak and small state in the IR image, the pixel resolution accounts for a small percentage. According to the definition of small targets by international standards, a target with fewer than $32 \times 32$ pixel values can be recognized as a small target, and most of the pixel values of drones in images are less than $32 \times 32$, which are typical of small infrared targets. Therefore, infrared small target detection based on a convolutional neural network (CNN) is a challenging problem.

Recently, strategies based on segmentation methods are attracting an increasing amount attention because of their ability to classify and locate objects at the pixel level [8]. Hao et al. proposed the use of BlendMask to extract more accurate instance segmentation features by fusing top-level and low-level semantic information with a more reasonable blender module [9]. Dai et al. proposed BoxInst, which only requires the box information of a target to predict the mask of a target [10]. Zhi et al. was proposed CondInst to ensure high accuracy without the assistance of a detector [11]. Daniel et al. proposed YOLACT++, which demonstrates detection speeds up to 33.5 fps [12]. Wang et al. also proposed SOLOv2, which was based on SOLO, and then proposed the idea of using dy-

namic learning to segment the target mask, decomposing it into two branches: the learning mask kernel and the generation mask [13]. He Keming et al. proposed Mask R-CNN, Mask R-CNN can effectively detect the target and output high quality instance segmentation mask [14]. YOLOv5 and YOLOv7 implemented instance segmentation by adding mask branches [15]. Pixel-level target detection based on instance segmentation is expected to suppress the influence of complex backgrounds and retain more infrared small target information [16,17]. However, in the process of feature extraction, approaches based on segmentation methods may still have the problem of target feature loss in the deep network, which can be attributed to the four factors described below.

The first factor is the large size of the convolutional step in the convolutional neural network. Target detection algorithms use convolutional neural networks as feature extraction tools; the feature map of the target continues to shrink during the convolution of the network [18,19], and the convolution step length is likely to be larger than the infrared small target size, which makes it difficult to transfer infrared small target features to the deep network [20–22]. The second factor is that the distribution of the dataset is not ideal. In the target detection datasets that are currently commonly used, such as MSCOCO [23], the number of samples of small targets accounts for a relatively small number, and the size difference between large and small targets is relatively large; thus, the detection algorithm has difficulty in adapting to changes in the scale of the target.

The third factor is the suboptimal hyperparameter setting of the a priori anchor. The sizes of infrared small targets often differ greatly from the set anchor box sizes, resulting in only a small portion of priori boxes overlapping with the true boxes, thus affecting detection performance. The fourth factor is the suboptimal setting of the intersection of union (IoU) threshold. The IoU between the candidate bounding box and ground truth box is small, the size of the IoU threshold directly affects the selection of positive and negative samples.

Currently, the means of improving the performance of infrared small target detection are mainly divided into four development directions.

(1)    Generative Adversarial Networks

A generative adversarial network is a deep generative model that learns to generate new data via adversarial training. A generator is used to generate an image after receiving random noise. A discriminator is used to discriminate whether an image is a real image in the dataset or a generator-generated image [24]. Li et al. proposed Perceptual to divide the images into those containing small and large targets, providing the first work to improve a small target detection algorithm based on GAN [25]. Kim et al. proposed a method that can generate synthetic training data for infrared small target detection [26]. Noh et al. modeled the target details by constantly adding new layers at the beginning of the training process when the output resolution of the Generator and Discriminator was low [27]. A method based on adversarial generative learning can effectively enhance the detail information of images, but there are two difficulties. Firstly, the loss function of a GAN in the training process is difficult to converge, leading to unstable experimental results. Secondly, during the training process, the generator generates limited samples, and the model is prone to collapse when the learning process stops, leading to errors in the final detection.

(2)    Multi-Scale Learning

The working theory of multi-scale learning is to use the details in the shallow convolution layer to help provide the location information of an infrared small target and to use the semantic information in the deep network to realize the classification of the target. However, the training of CNN models based on feature pyramids has very high computing power and memory requirements. In order to reduce computational resources and obtain better feature fusion, Lin et al. created a feature pyramid with strong semantic features at all levels using only one scale of input [28]. Cui et al. proposed a jump connection to fuse more scale features for the problem that the feature information of small targets is still incomplete after multiple samplings [29].

(3)    Use of Feature Context Information

Infrared small targets account for a very small proportion of an image, and the information obtained from the local area of the image is very limited. In addition, infrared small target detectors usually ignore the contextual features outside the local area. Therefore, researchers have proposed a detection method based on contextual information by using the relationships between small targets and other targets or the background. Zagoruyko et al. used the region cropping method to crop four different multiples of regions at the center of the original region proposal and then performed region of interest pooling to cascade the pooled information together to achieve the effect of fusing contextual information [30]. Guan et al. constructed context-aware features using the pyramidal pooling of multilayer context information [31]. Hu et al. designed context-aware target region pooling by adding the proportion of context information to the feature map to avoid losing small target information [32].

(4)    Improvement of Loss Function

The loss function is a means of measuring the gap between the predicted and actual values of the output of a neural network. In a neural-network-based target detection task, a metric called the intersection of union (IoU) is commonly used. The IoU can describe the relationship between the prediction frame and the real frame well, but the value of the IoU will be zero if the two target frames do not overlap [33]. Hamid et al. proposed the GIoU, which solves the problem that when the two target boxes do not intersect, the gradient is zero and cannot train the network [34]. Zheng et al. proposed the DIoU, which considers the distance between the target and the prediction, overlap rate and scale to make the target box regression more stable [35]. The authors of the DIOU also proposed the CIOU, which takes into account the aspect ratios of the boxes [35]. Zhang et al. proposed EIOU loss and added focal to focus on high-quality predictor boxes, introducing focal loss to optimize the sample imbalance in the bounding box regression task so that the regression process focuses on high-quality anchor boxes [36].

In short, detecting small drones in infrared images is challenging. The traditional target detection algorithm cannot adapt to the complex scene, especially when the drone target scale is small and the background noise is complex. The target detection method based on CNN cannot accurately detect the infrared small target. Pixel-level target detection based on instance segmentation is expected to suppress the influence of a complex background and retain more infrared small target information [16,17]. However, in the process of feature extraction, there may still be the problem of target feature loss. To solve this problem, we investigated the causes of infrared small target feature loss in the deep network and the corresponding solutions.

Our research contributions will be detailed later and can be summarized as follows:

(1)    We added an infrared small target detection module (IRSTDM) to effectively realize the extraction and retention of infrared small target information based on the current advanced segmentation detector YOLOv5.

(2)    We introduced normalized Wasserstein distance (NWD) to optimize the boundary frame loss function, aiming to solve the problem that the calculated loss function based on the IOU is sensitive to the position of small targets.

(3)    We built and published a new SIDD dataset, carried out pixel-level annotation on infrared drone images in the dataset and published all masks used for segmentation.

(4)    We conducted a large number of experiments between the proposed IRSDD-YOLOv5 and eight mainstream segmentation detection methods in the SIDD dataset and verified that the proposed method is superior to the most advanced method.

The rest of this article is as follows. In Section 2, we introduce the construction of the data set and describe in detail the IRSDD-YOLOv5 structure and improved methods. In Section 3, we introduce the details of the experiment and analyze the obtained results, describing prospective work for future research. We summarize the entire research work in Section 4.

## 2. Materials and Methods

### 2.1. Dataset

Currently, in commonly used target detection datasets in which the sample number of small targets is relatively small, such as MSCOCO, the size differences between large targets and small targets are relatively large, and the lack of drone targets brings certain difficulties for the network in adapting to the target [23]. In addition, the number of small targets, such as a tiny person, in a specific small target dataset may be small in one image. If the number of small targets is less than 20, it may be greater, such as more than 100 [37]. The different densities of small targets makes it difficult to use a unified method to improve the effect of detecting of small targets.

Current datasets dedicated to detecting small drone targets based on IR images are mainly the ground/air background infrared image weak aircraft target detection and tracking dataset and the first CVPR anti-drone dataset. The ground/air context infrared image weak aircraft target detection and tracking dataset is oriented to low-flying weak aircraft target detection and tracking applications, and it provides a set of algorithm test dataset with one or more fixed-wing drone targets as detection objects by field photography and data preparation processing. The scenes in the datasetcover the sky, the ground and a variety of backgrounds, with a total of 22 data segments, 30 traces, 16,177 frames of images, and 16,944 targets; however, the targets in these dataset scenes are fixed-wing drones, while in civilian black flights, most drones are quadrotor drones [38]. CVPR's first anti-drone dataset provides a large amount of drone flight footage, but the data are based on videos of drones with location information, without pixel-level annotation and without subdividing the flight scenes of the drones. We proposed the SIDD dataset, which distinguishes four scenarios in which drones may invade, provides pixel-level annotation of the targets and is published at https://github.com/Dang-zy/SIDD.git (accessed on 27 April 2023).

The infrared drone target in the dataset image is a quadrotor drone, which is a typical low and slow small target. In infrared images with complex backgrounds, the edge information of the target is also difficult to describe clearly, and the interference sources are different for different scenes. In order to restore the real intrusion scene of the drone as much as possible, we distinguished between four scenes to explore the impacts of different backgrounds on drone detection. The SIDD dataset contains 4737 images of $640 \times 512$ pixels, including 2151 images of mountain scenes, 1093 images of city scenes, 780 images of sky scenes, and 713 images of sea scenes, Figure 2 shows an example of the SIDD dataset. We divided 80% of the dataset for training and 20% for testing. In this study, we conducted extensive experiments on the SIDD dataset using mainstream segmentation algorithms.

### 2.2. Proposed Method

In this section, we provide a general introduction to the proposed IRSDD-YOLOv5, and the following subsections describe the overall structure and main innovations of the proposed network model.

According to the width and depth of the network, YOLOv5 is divided into four different hierarchical models, which are model s, model m, model l and model x. YOLOv5 is the network with the smallest depth and the smallest width of the feature graph in the series. According to the target characteristics and practical application requirements of the detection of infrared small drones, our segmentation network is model s. IRSDD-YOLOv5's instance segmentation process is to create weighted summations between the prototype mask outputted at the tail of the neck network and the mask coefficient outputted by the prediction head so as to obtain the instance segmentation result. As shown in Figure 3, the backbone network and neck network extract the feature information of the input image and send it into the prediction head to output the mask coefficient. After NMS suppression, the unnecessary coefficient can be removed. At the tail of the neck network, the prototype of the target mask can be outputted, and it is weighted and summed with the mask coefficient

after NMS suppression to obtain the target mask. After the threshold suppression, the mask whose confidence is greater than the threshold value is finally displayed on the image.

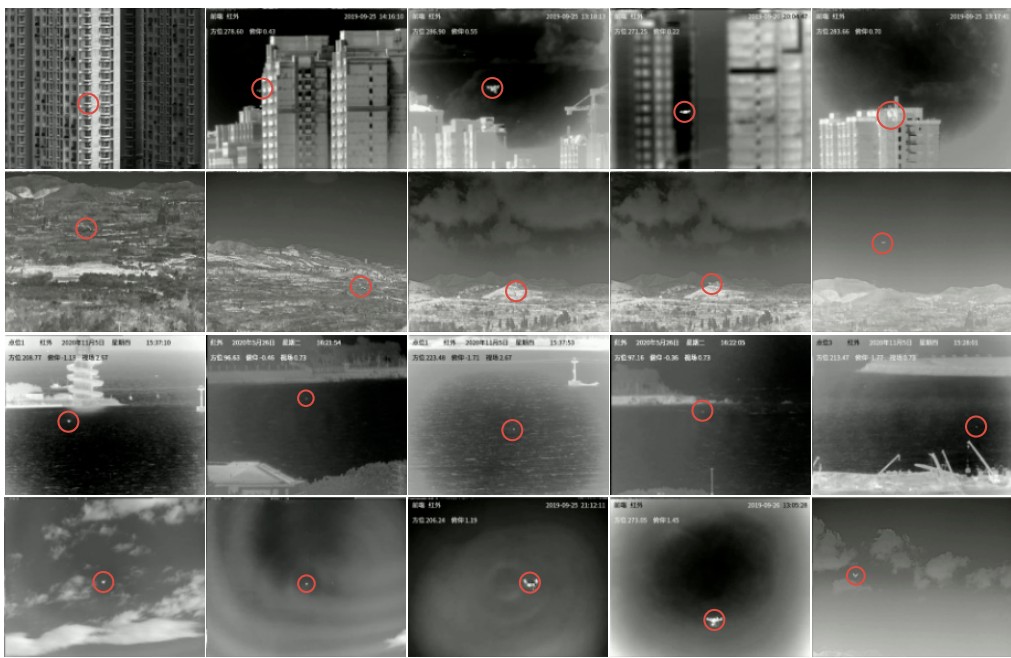

**Figure 2.** Examples of SIDD dataset, from top to bottom: city scene, mountain scene, sponge scene and sky background. The drone targets in the images are marked with red circles.

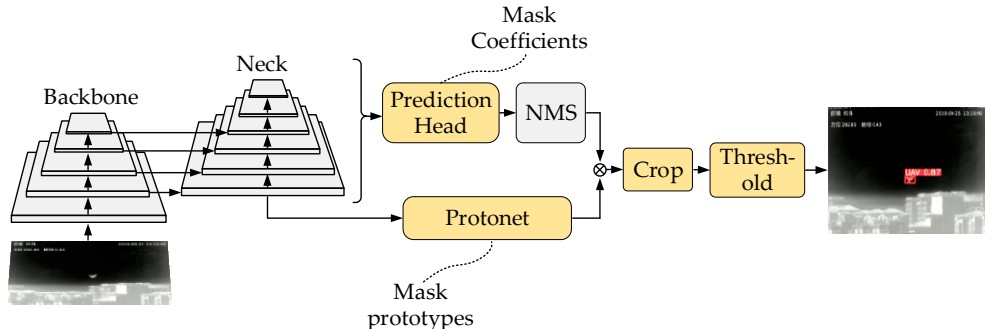

**Figure 3.** Segmentation process of IRSDD-YOLOv5 network.

### 2.2.1. Overall Architecture

The network structure of IRSDD-YOLOv5 is shown in Figure 4. It consists of two parts: feature extraction and feature fusion prediction.

In order to provide a solution for the difficulty of detecting infrared small drones, we added an infrared small target detection module (IRSTDM) to the neck network in the original YOLOv5 model so as to make full use of the global contextual information. A prediction head for small target detection (the first prediction head) was also added to constitute a new infrared small target detection layer. The new prediction head and the other three prediction heads form a four-prediction-head structure that can mitigate the negative impact of detecting changes on the scale of infrared small targets.

IRSDD-YOLOv5 contains a total of four detection heads for detecting tiny, small, medium and large objects. The C3 module and the Space Pyramid Pool Network (SPPF) module are used as the backbone network of IRSDD-YOLOv5. The specific structures of the C3 and SPPF modules are shown in Figure 5. At the same time, we also introduce dNWD to optimize the positioning loss function twice, appropriately reduce the sensitivity to small

targets and reduce the false alarm rate. Compared to YOLOv5, our IRSDD-YOLOv5 can handle captured drone images better.

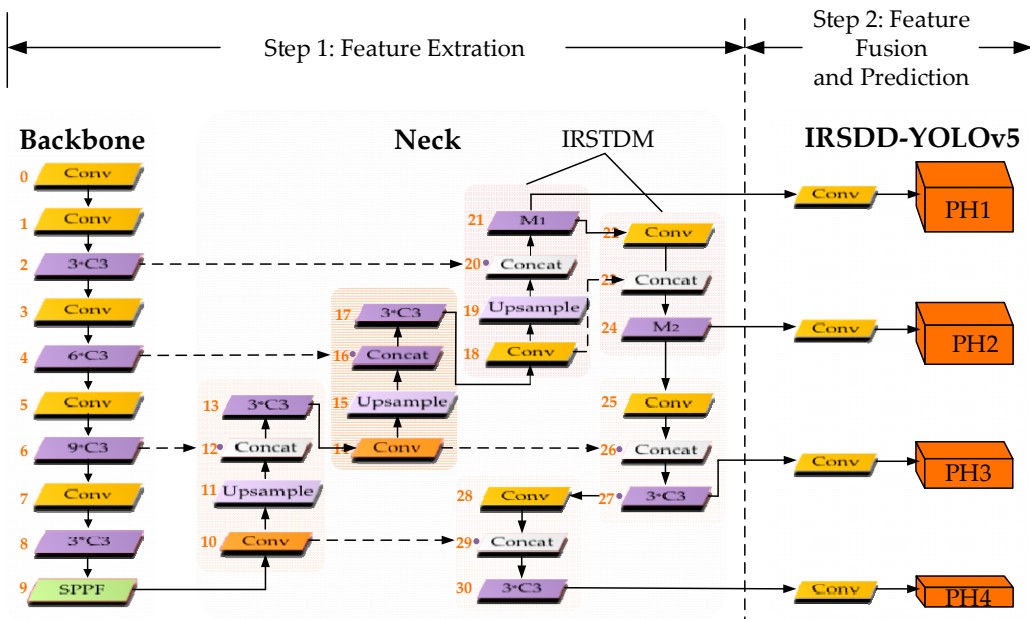

**Figure 4.** Overall architecture of IRSDD-YOLOv5. A PANet-like structure is used in the neck network, and the red part is the infrared small drone detection module added to the neck network. The four prediction heads use the feature maps generated from the neck network to fuse information about the targets. In addition, the number of each module is marked with an orange number on the left side of the module.

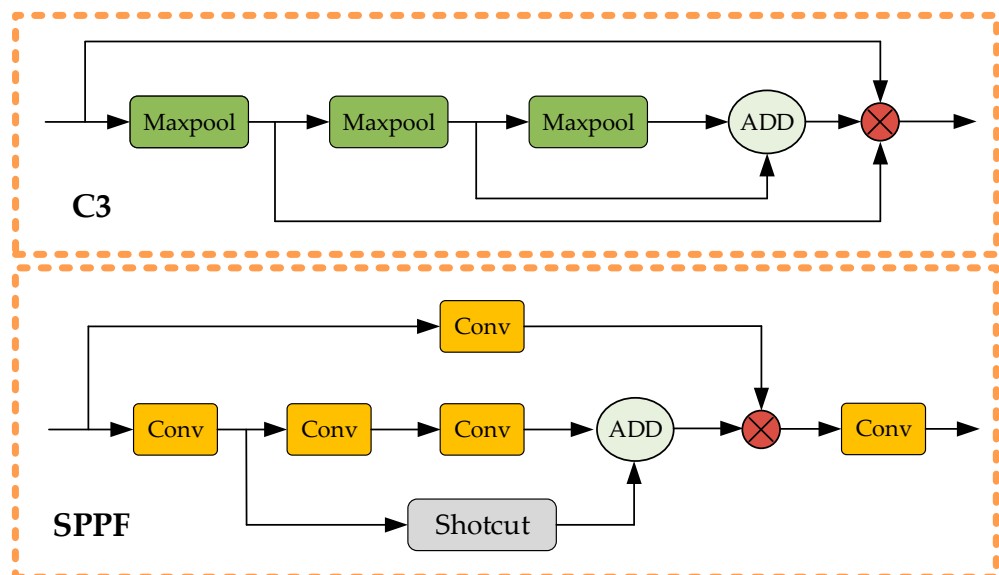

**Figure 5.** The specific structure of C3 and SPPF modules.

### 2.2.2. Infrared Small Drone Detection Module

Deep feature networks extract the semantic information of small targets; however, they may risk losing spatial details of the infrared small targets. Shallow feature networks retain spatial location information, but they lack a deep semantic understanding of the target. The feature representation of infrared small drones is difficult when deep learning models do not fully utilize the information from different feature layers. Xu et al. demonstrated

that by concatenating the characteristic information output from the shallow network and the previous shallow network,, more small target information and edge information can be retained [39]. Inspired by Xu et al., we constructed a hierarchical contextual IRSTDM in the intermediate feature layer of the neck network to make full use of the global information. The module structure is shown in Figure 6.

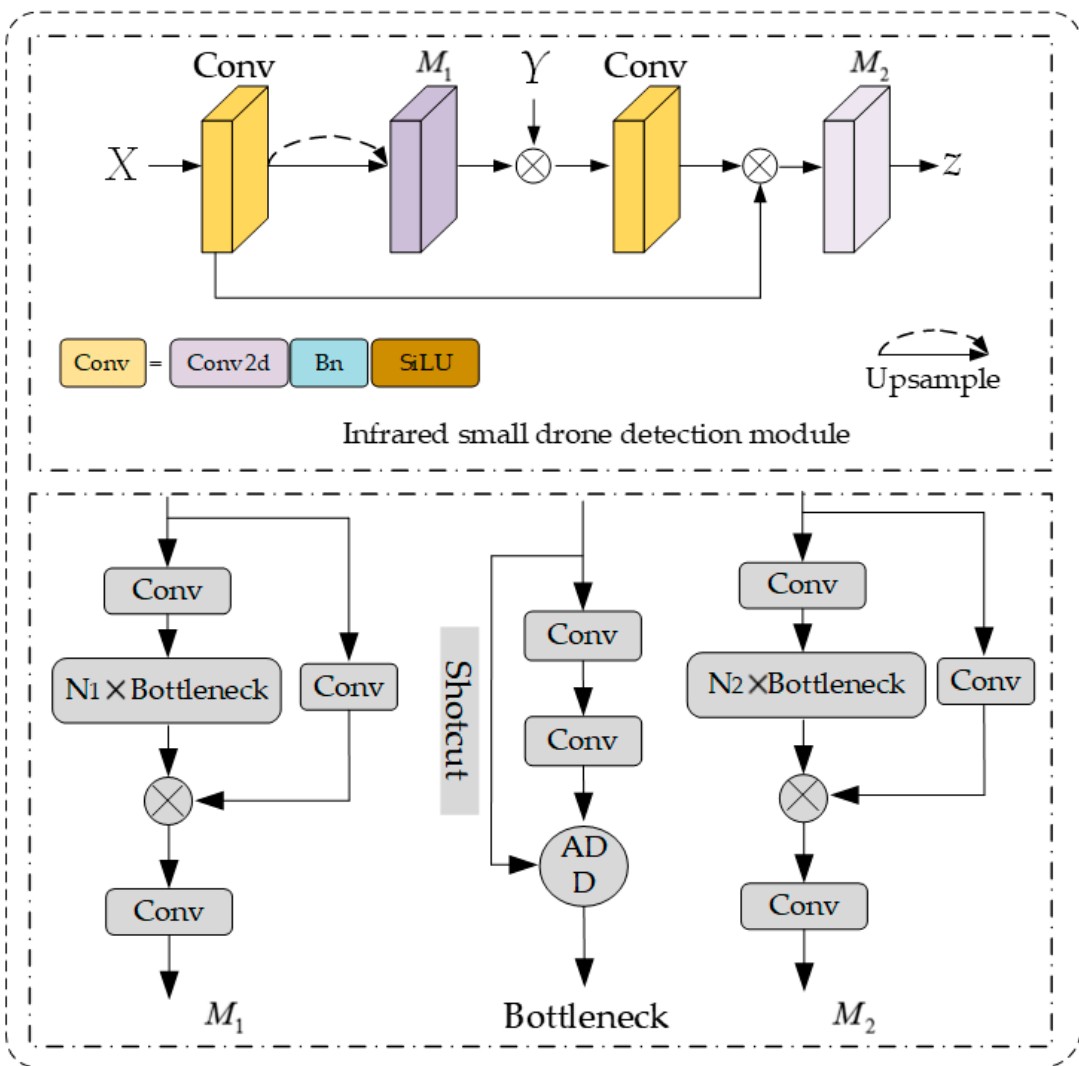

**Figure 6.** The top part of the figure shows how IRSTDM works, and the bottom part shows the detailed part of the module.

Our proposed infrared small target detection module (IRSTDM) can not only extract rich semantic information of infrared small targets but can also maintain the depth feature representations of infrared small targets and avoid the situation that infrared small targets may be lost in the deep network. IRSTDM also further solves the problems of information redundancy and insufficient feature fusion between different network layers. The calculation method of IRSDDM using prior global context information is as follows:

$$H = M_1(\delta(Conv(X))) \otimes Y \tag{1}$$

$$Z = M_2(Conv(H) \otimes Conv(X)) \tag{2}$$

where $X$ and $Y$ denote the semantic information from the deep network and the detailed information from the shallow network. $\delta$ denotes upsampling, and $\otimes$ indicates the cascading

of different inputs. After convolution, the shallow semantic feature graph $X$ is up-sampled, and after processing by the $M_1$ module, it is cascaded with the corresponding deep feature graph $Y$ to obtain $H$. $H$ is then cascaded with the output of $X$ after convolution to obtain the feature map and then processed by $M_2$ to output $Z$.

$M_1$ and $M_2$: Features can still be effectively activated when infrared small target information is lost, and distinguishable features can be captured through multiple convolution branch channels to replace standard empty convolution to reduce model complexity while ensuring model accuracy. The calculation method is as follows:

$$M_j = N_i \times B(Conv(x)) \otimes Conv(x) \quad i = 1, 2, 3, 4 \quad j = 1, 2 \tag{3}$$

where $\otimes$ indicates the cascading of different inputs, $B$ denotes the number of Bottleneck modules and $x$ represents the input of the $M_j$ module.

Regarding Bottleneck, generally, increasing the depth of the network can improve the accuracy, but this will increase the amount of computation required, while Bottleneck can increase the depth and save the computation, which is calculated by:

$$O = Conv(Conv(x)) \oplus S(x) \tag{4}$$

where $\otimes$ denotes the cascading of different inputs, $x$ and $O$ represent the input and output representatives of the Bottleneck modules and $S$ stands for the weighted shortcut function.

### 2.2.3. Optimization of Loss Function

The loss function in the YOLOv5-seg network model consists of $L_{\text{class loss}}$, $L_{\text{box loss}}$, $L_{\text{seg loss}}$ and $L_{\text{object loss}}$, where $L_{\text{class loss}}$ and $L_{\text{box loss}}$ are the classification loss and the bounding box loss, respectively, and $L_{\text{seg loss}}$ and $L_{\text{object loss}}$ are the segmentation loss and the confidence loss.

Both the confidence loss and classification loss are calculated using the BCE with Logits Loss function. The segmentation loss is calculated by the single_mask_loss function to calculate the true mask value and the predicted mask value, while the bounding box and segmentation loss are calculated by CIoU using the following formula:

$$L_{box\ IoU} = 1 - L_{IoU} + \frac{p^2(b, b_{gt})}{c^2} + \alpha v \tag{5}$$

In Equation (5), $b$ denotes the prediction frame and $b_{gt}$ denotes the true box. $c$ denotes the diagonal distance of the smallest closed area that can contain both the prediction box and the true box, $\alpha$ is the balance parameter and $v$ is used to measure whether the aspect ratio is consistent.

$$v = \frac{4}{\pi^2} \left( \arctan \frac{w_{gt}}{h_{gt}} - \arctan \frac{w}{h} \right)^2 \tag{6}$$

$$\alpha = \frac{v}{(1 - L_{IoU}) + v} \tag{7}$$

As can be seen from Equation (6), when the aspect ratio of the predicted box is as large as that of the real frame, the value of $v$ is zero. At this time, the penalty term of the aspect ratio does not play a role, and the CIoU loss function does not achieve a stable work.

To solve the above problem, we introduced a new loss function, NWD, by analyzing the target characteristics of an infrared small drone. There are often some background pixels in the bounding box of the infrared drone target, and in these bounding boxes, the foreground and background pixels are concentrated on the center and the boundary of the bounding box, respectively. In order to better describe the weights of different pixels in the bounding box, the bounding box can be modeled as a two-dimensional Gaussian distribution in which the center pixel of the bounding box has the highest weight, and the importance of the pixel decreases from the center to the boundary. Figure 7 shows

the calculation process of NWD. Deviation is the deviation of the center pixel between two boxes.

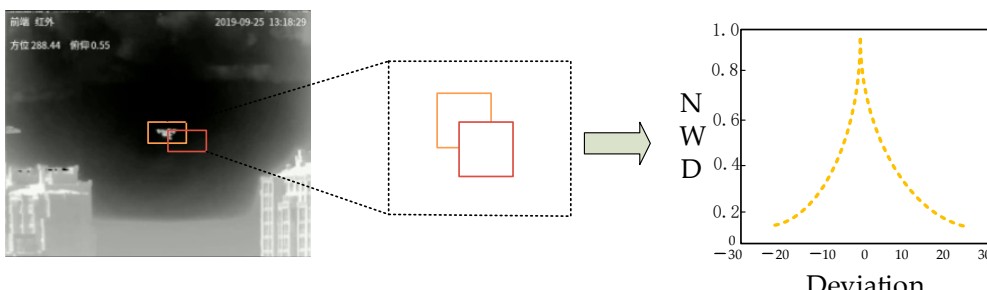

**Figure 7.** Example of a procedure for calculating the NWD between two boxes.

The following shows the detailed NWD calculation process:

The similarity between the true bounding box a and the predicted bounding box b can be translated into the Wasserstein distance between the two Gaussian distributions as:

$$W_2^2(N_a, N_b) = ||([cx_a, cy_a, \frac{w_a}{2}, \frac{h_a}{2}]^T, [cx_b, cy_b, \frac{w_b}{2}, \frac{h_b}{2}]^T)||_2^2 \tag{8}$$

where $cx$ and $cy$ are the central coordinate points of the bounding box, and $w$ and $h$ are the width and height of the bounding box.

However, $W_2^2(N_a, N_b)$ is a distance metric and cannot be used directly as a similarity metric. Therefore, we use its exponential form, normalized as IoU, to obtain a new metric called the normalized Wasserstein distance (NWD).

$$NWD(N_a, N_b) = \exp(-\frac{\sqrt{W_2^2(N_a, N_b)}}{C}) \tag{9}$$

In this paper, we designed the bounding box loss function by analyzing the advantages and disadvantages of different loss functions and optimizing the loss function by combining the CIOU and NWD.

$$L_{NWD} = 1 - NWD(N_{\rm p}, N_g) \tag{10}$$

$$L_{\rm box\ loss} = (1 - \beta)L_{NWD} + \beta(1 - L_{CIOU}) \tag{11}$$

where $N_p$ is the Gaussian distribution model of the predicted enclosing frame and $N_g$ is the Gaussian distribution model of the real enclosing frame. $(1 - \beta)$ are the proportions occupied by $L_{NWD}$.

## 3. Results and Discussion

In this section, we introduce qualitative and quantitative evaluations of the proposed method using a self-made single-frame infrared small target drone dataset (SIDD) and analyze the detection results. Firstly, Section 3.1 describes the evaluation indexes of algorithm performance, and Section 3.2 describes the details of the computer hardware configuration and experiment implementation. Subsequently, in Section 3.3, qualitative and quantitative comparisons are made between IRSDD-YOLOv5 and the current most advanced instance segmentation methods, and the reasons for the different accuracies of drone detection achieved in different scenarios are analyzed. We present detailed results of the ablation study in Section 3.4. Finally, in Section 3.5, we present certain expectations for future research work based on current drone prevention and control technologies.

### 3.1. Evaluation Metrics

We consider drone detection a pixel-level instance segmentation task. Therefore, we compare the performances of different algorithms using classical instance segmentation

evaluation metrics to assess the abilities of different algorithms to accurately localize and characterize the shape of small infrared targets to ensure that the network detects the target and ensures as few false positives as possible.

(1)  Average precision (AP): The main algorithm evaluation metrics include $AP_{\{0.5:0.95\}}$, $AP_{50}$ and $AP_s$. $AP_{\{0.5:0.95\}}$ is the average value for an IoU of {0.5, 0.55, ----, 0.95}, and AP50 is the value taken for an IoU threshold of 0.5. $AP_s$ is the AP value for small targets (less than $32 \times 32$ pixel values).

(2)  Parameters (Para): The number of parameters refers to the value of the parameters included in the model. Each number corresponding to the weight matrix in the convolution and the fully connected layers used in the model is part of the number of parameters. The number of parameters is the key to the machine learning algorithm. The size of the model parameters reflects the complexity of the model to some extent.

(3)  Frames per second (FPS): The higher the FPS value, the better is the real-time processing ability of the model when running the detection algorithm under the same hardware conditions. FPS = 1/latency, where latency is the time taken by the network to predict an image.

### 3.2. Implementation Details

We used PyTorch 1.8 and TorchVision 0.9 to implement the proposed model on a computer with a single GPU. The computer's processor was the AMD Ryzen 7 5800H, the GPU was an RTX3060, the initial learning rate of the network was set to 0.0025, the weight attenuation was set to 0.005, the batch size was set to 2 and an AdaGrad training optimizer was used. All images were adjusted to $640 \times 640$ pixels before being fed into the network and were normalized to speed up network convergence. All CNN-based instance segmentation models were trained on SIDD datasets for 50 cycles.

### 3.3. Comparison with the Latest Methods

To verify the superiority of IRSDD-YOLOv5, we compared it with several state-of-the-art methods: a top-down meets bottom-up segmentation network (Blendmask) [9], high-performance instance segmentation using only the box's annotation (BoxInst) [10], a segmentation network using dynamic mask headers (CondInst) [11], Yolact++ [12], a segmentation network with a double-branch output generation mask (Solov2) [13], the generation of a high-quality segmentation mask network for each target instance (Mask-Rcnn) [14] and the You Only Lock Once series (Yolov5 and Yolov7) [15]. Traditional methods (top-hat filtering and infrared patch-image modeling) were not considered due to their inability to be trained by the dataset and their poor performance [1,2]. To ensure the completeness of the experiments, we compared quantitatively and qualitatively the detection results of different scenes in the SIDD dataset.

#### 3.3.1. Quantitative Results

Tables 1–4 lists the quantization results of IRSDD-YOLOv5 and other advanced segmentation detection methods in different scenarios of SIDD dataset. *AP* represents the detection accuracy, para(M) represents the complexity of each different model and the FPS calculation value describes the real-time computation of each of the different models. In the tables, the best results for each column are highlighted in orange bold, the second best in blue bold and the third best in green bold. The -- in each column represents a zero result or a null output for an indicator of the algorithm. The detection results of our proposed method in different scenes from the SIDD dataset are highlighted with grey shading. From Tables 1–4, it can be seen that IRSDD-YOLOv5 obtained excellent detection results in different scenes from the SIDD dataset, significantly improving the accuracy of detecting an infrared small drone.

**Table 1.** Performances of different algorithms for drone detection based on single-frame infrared images in urban scenarios.

| Methods | City Scene | | | | |
|---|---|---|---|---|---|
| | $AP_{\{0.5:0.95\}}$ | $AP_{50}$ | $AP_s$ | Para(M) | FPS |
| Blendmask | 0.621 | 0.940 | 0.613 | 287.6 | 8.2 |
| BoxInst | 0.197 | 0.538 | 0.197 | 273.8 | 9.6 |
| CondInst | 0.565 | 0.936 | 0.564 | 272.6 | 9.6 |
| Solov2 | 0.620 | 0.936 | 0.607 | 372.0 | 8.6 |
| Mask-Rcnn | 0.629 | 0.937 | 0.621 | 353.3 | 3.97 |
| Yolov5 | 0.477 | 0.929 | 0.469 | 15.2 | 29.68 |
| Yolov7 | 0.440 | 0.877 | 0.435 | 76.3 | 16.28 |
| Yolact++ | 0.423 | 0.902 | -- | 199.0 | 10.77 |
| ours | 0.473 | 0.939 | 0.469 | 15.9 | 25.91 |

**Table 2.** Performances of different algorithms for drone detection based on single-frame infrared images in mountainous scenarios.

| Methods | Mountain Scene | | | | |
|---|---|---|---|---|---|
| | $AP_{\{0.5:0.95\}}$ | $AP_{50}$ | $AP_s$ | Para | FPS |
| Blendmask | 0.423 | 0.775 | 0.423 | 287.6 | 8.99 |
| BoxInst | -- | 0.013 | -- | 273.8 | 10.26 |
| CondInst | 0.284 | 0.731 | 0.284 | 272.6 | 10.35 |
| Solov2 | -- | -- | -- | 372.0 | 9.06 |
| Mask-Rcnn | 0.416 | 0.749 | 0.416 | 353.3 | 4.04 |
| Yolov5 | 0.278 | 0.760 | 0.278 | 15.2 | 40.08 |
| Yolov7 | 0.269 | 0.746 | 0.269 | 76.3 | 20.79 |
| Yolact++ | 0.177 | 0.625 | -- | 199.0 | 11.93 |
| ours | 0.277 | 0.798 | 0.277 | 15.9 | 29.03 |

**Table 3.** Performances of different algorithms for drone detection based on single-frame infrared images in sea surface scenarios.

| Methods | Sea Surface Scene | | | | |
|---|---|---|---|---|---|
| | $AP_{\{0.5:0.95\}}$ | $AP_{50}$ | $AP_s$ | Para | FPS |
| Blendmask | 0.455 | 0.842 | 0.456 | 287.6 | 7.83 |
| BoxInst | -- | -- | -- | 273.8 | 8.97 |
| CondInst | 0.292 | 0.819 | 0.292 | 272.6 | 8.86 |
| Solov2 | -- | -- | -- | 372.0 | 8.08 |
| Mask-Rcnn | 0.463 | 0.933 | 0.463 | 353.3 | 4.04 |
| Yolov5 | 0.334 | 0.894 | 0.334 | 15.2 | 20.64 |
| Yolov7 | 0.355 | 0.930 | 0.355 | 76.3 | 13.82 |
| Yolact++ | 0.163 | 0.445 | -- | 199.0 | 9.42 |
| ours | 0.375 | 0.934 | 0.375 | 15.9 | 16.01 |

**Table 4.** Performances of different algorithms for drone detection based on single-frame infrared images in sky scenarios.

| Methods | Sky Background | | | | |
|---|---|---|---|---|---|
| | $AP_{\{0.5:0.95\}}$ | $AP_{50}$ | $AP_s$ | Para | FPS |
| Blendmask | 0.725 | 0.986 | 0.717 | 287.6 | 7.81 |
| BoxInst | 0.395 | 0.806 | 0.397 | 273.8 | 9.06 |
| CondInst | 0.673 | 0.977 | 0.648 | 272.6 | 8.91 |
| Solov2 | 0.686 | 0.934 | 0.664 | 372.0 | 7.95 |
| Mask-Rcnn | 0.711 | 0.987 | 0.703 | 351.3 | 4.03 |
| Yolov5 | 0.592 | 0.977 | 0.570 | 15.2 | 21.68 |
| Yolov7 | 0.580 | 0.974 | 0.561 | 76.3 | 14.61 |
| Yolact++ | 0.561 | 0.958 | -- | 199.0 | 9.74 |
| ours | 0.593 | 0.977 | 0.578 | 15.9 | 16.43 |

As1 shown in Tables 1 and 2, the $AP_{\{0.5:0.95\}}$ and $AP_{50}$ values of IRSDD-YOLOv5 in urban scenes reached 47.3% and 93.9%, respectively, and the AP50 value is the second highest, only 0.01% lower than that of the first-ranked Blendmask. However, in terms of real-time performance, our algorithm is several orders of magnitude higher than Blendmask, ranking second among all algorithms. The $AP_{\{0.5:0.95\}}$ and $AP_{50}$ measurements of IRSDD-YOLOv5 in mountain scenarios reached 27.7% and 79.8%, respectively. Compared with other algorithms, $AP_{50}$ ranked first and reached a peak value of 79.8%, 3.8% higher than YOLOv5. However, in the process of training data on the mountain scene, the loss function value of the SOLOv2 algorithm did not converge because the background in the mountain scene is relatively complex. Therefore, this algorithm provides no result for the mountain scene.

As shown in Tables 3 and 4, in the sea surface scenario, the $AP_{\{0.5:0.95\}}$ and $AP_{50}$ values of IRSDD-YOLOv5 reached 37.5% and 93.4%, respectively. The $AP_{50}$ value of IRSDD-YOLOv5 in the sea surface scenario reached 93.4%, ranking the first among all algorithms and 4% higher than the $AP_{50}$ value of YOLOv5. Its real-time performance ranked second. In the process of training on the sea surface scene data, the target in the sea surface scene is generally too small. As a result, the value of the loss function of the SOLOv2 algorithm did not converge, and the detection accuracy of BoxInst is close to 0. Therefore, these two algorithms provide no results for the sea surface scene. In the sky scene, due to the simple background, most detection methods achieved excellent detection results. However, the IRSDD-YOLOv5 proposed by us demonstrated faster real-time performance and fewer parameters, ranking second in terms of real-time performance.

In general, we adopted IRSDD-YOLOv5 by adding a small target detection layer and introducing the NWD to optimize the boundary frame loss function, which is improved in all scenarios when compared with YOLOv5. Due to the addition of the small target detection layer, the network complexity of IRSDD-YOLOv5 is increased. Compared with YOLOv5, the number of parameters is slightly increased, and the real-time performance is somewhat decreased. However, under the condition of obtaining the same accuracy, compared with other mainstream segmentation algorithms, the IRSDD-YOLOv5 algorithm proposed by us demonstrates greater real-time performance and detection accuracy. It is worth noting that the real-time performance of two-stage network-based methods (such as Mask-Rcnn) is poor, less than 5 FPS/s, because a two-stage algorithm usually first uses a selective search algorithm to locate the candidate regions in the image and then uses a classifier to classify each candidate region. These two steps require a large amount of computing resources, so the efficiency of a two-stage algorithm is relatively low.

### 3.3.2. Qualitative Results

Figure 8 shows the qualitative results obtained using the proposed IRSDD-YOLOv5 algorithm and other methods in four scenes with different backgrounds on the SIDD dataset. In the sea scenario, Blendmask and BoxInst have missed and false detections for the target, CondInst has a missed detection for the target; in the mountain scenario, Mask-Rcnn has a false detection, Yolact++ has a missed detection for the target. It is worth noting that the segmentation results of IRSDD-YOLOv5 have no errors, which indicates that it shows good detection performance for small targets in complex backgrounds (such as mountains and the sea surface).

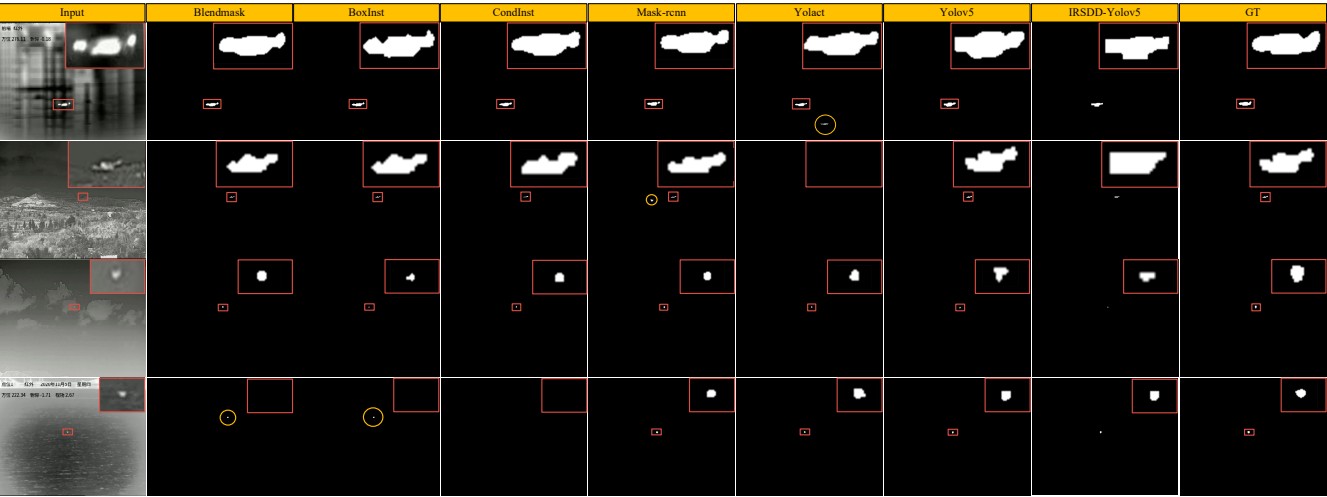

**Figure 8.** Visualization of target detection results. The leftmost is the input map, and from left to right is the mask map of target results of mainstream segmentation methods. The target area marked by red circle is enlarged in the upper right corner. GT represents the real area region of the target mask.

Figure 8 shows the segmentation results of IRSDD-YOLOv5 and other methods in four scenarios with different backgrounds on the SIDD dataset. As the detected target was too small, we kept the target mask areas of the segmentation results in the figure, and the background parts are represented in black area so that the instance segmentation results can be more clearly displayed. In the mountain scene, Mask-Rcnn misdetected the target, while Yolact++ failed to detect the target. In the sea surface scene, Blendmask and BoxInst have an omission and a false detection for the target, while CondInst has an omission for the target. It is worth noting that the segmentation result of IRSDD-YOLOv5 has no error, which indicates that it has good detection performance for small targets in complex background (such as mountains and the sea surface) and is more robust to these scenario changes.

Figures 9 and 10 show the three-dimensional visualization detection results of different detection methods in different scenes in the SIDD dataset, with the images missed by this algorithm containing a blue plane. It can be seen from the synthesis of Figures 9 and 10 that the scale of the drone in the urban scene is large, while that of the drone in the sea scene is the smallest. For targets with relatively large scales, although the current mainstream segmentation methods have achieved good detection performance, there are still false detections for small-scale targets. Because there's no false detections, YOLOv5 was superior to the other segmentation methods in terms of its detection accuracy, but IRSDD-YOLOv5 obtained a position distribution more consistent with the real region of the target, and its detection results were more accurate.

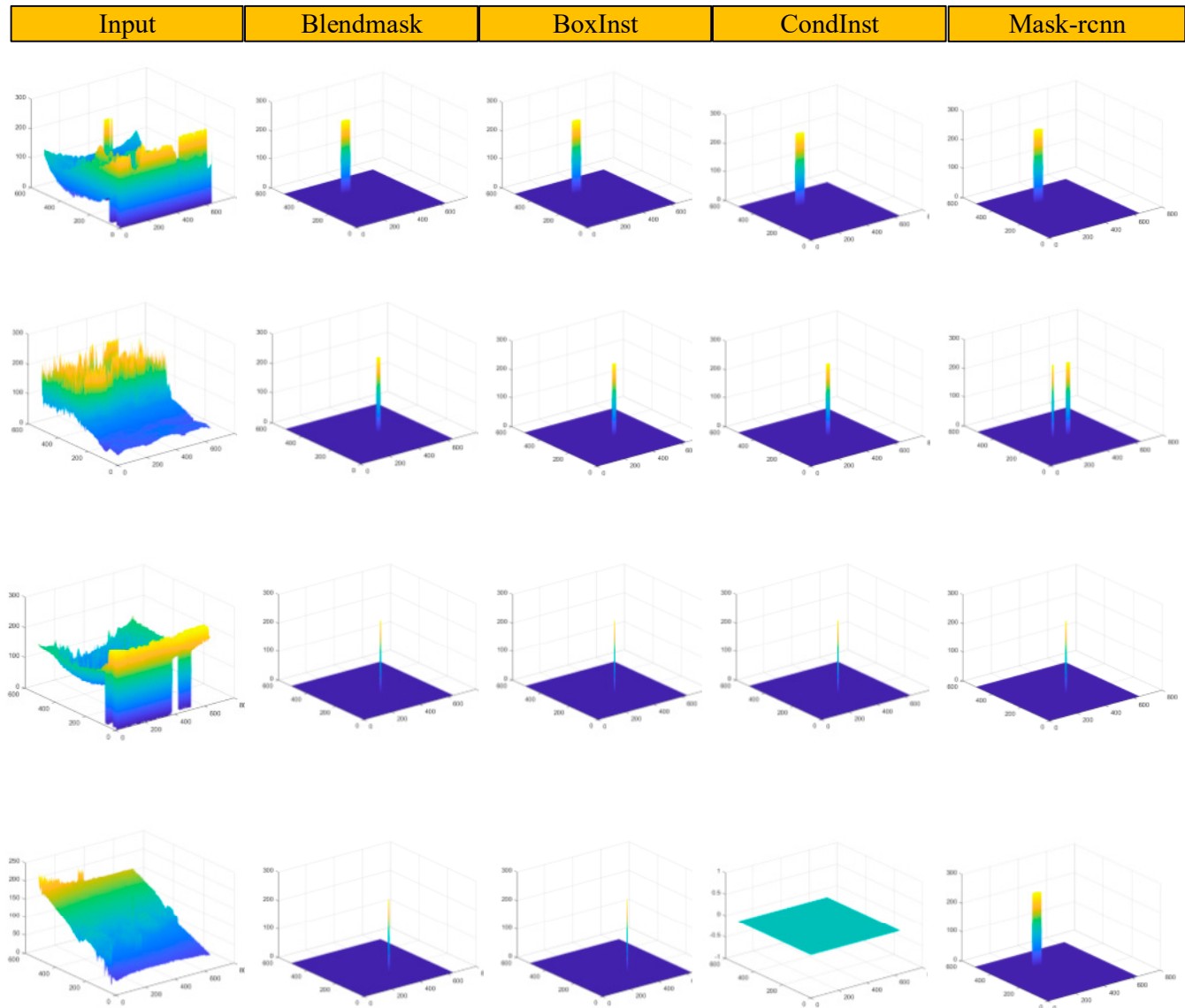

**Figure 9.** Three-dimensional visualization qualitative results of different instance segmentation methods. From left to right is the input original image, the segmentation result of BLendmask, BoxInst, CondInst, Maskrcnn. From top to bottom are the city scene, mountain scene, sea scene and sky scene.

In summary, the qualitative and quantitative results show that the scales of the drone targets in the SIDD datasets vary greatly, the backgrounds are complex, the shapes and sizes of targets vary and many targets have unclear boundaries. However, IRSDD-YOLOv5 can obtain a higher detection accuracy with a lower parameter number and a higher real-time performance and can accurately detect targets in different scenarios and different scales, indicating that the detection performance of IRSDD-YOLOv5 is robust and that our algorithm can more accurately detect complex scenarios and multi-scale changing targets. Our proposed method of adding a small target detection layer and improving the loss function not only obtains clear target boundary information but also maintains and integrates sufficient infrared drone target image context information so that IRSDD-YOLOv5 has excellent performance.

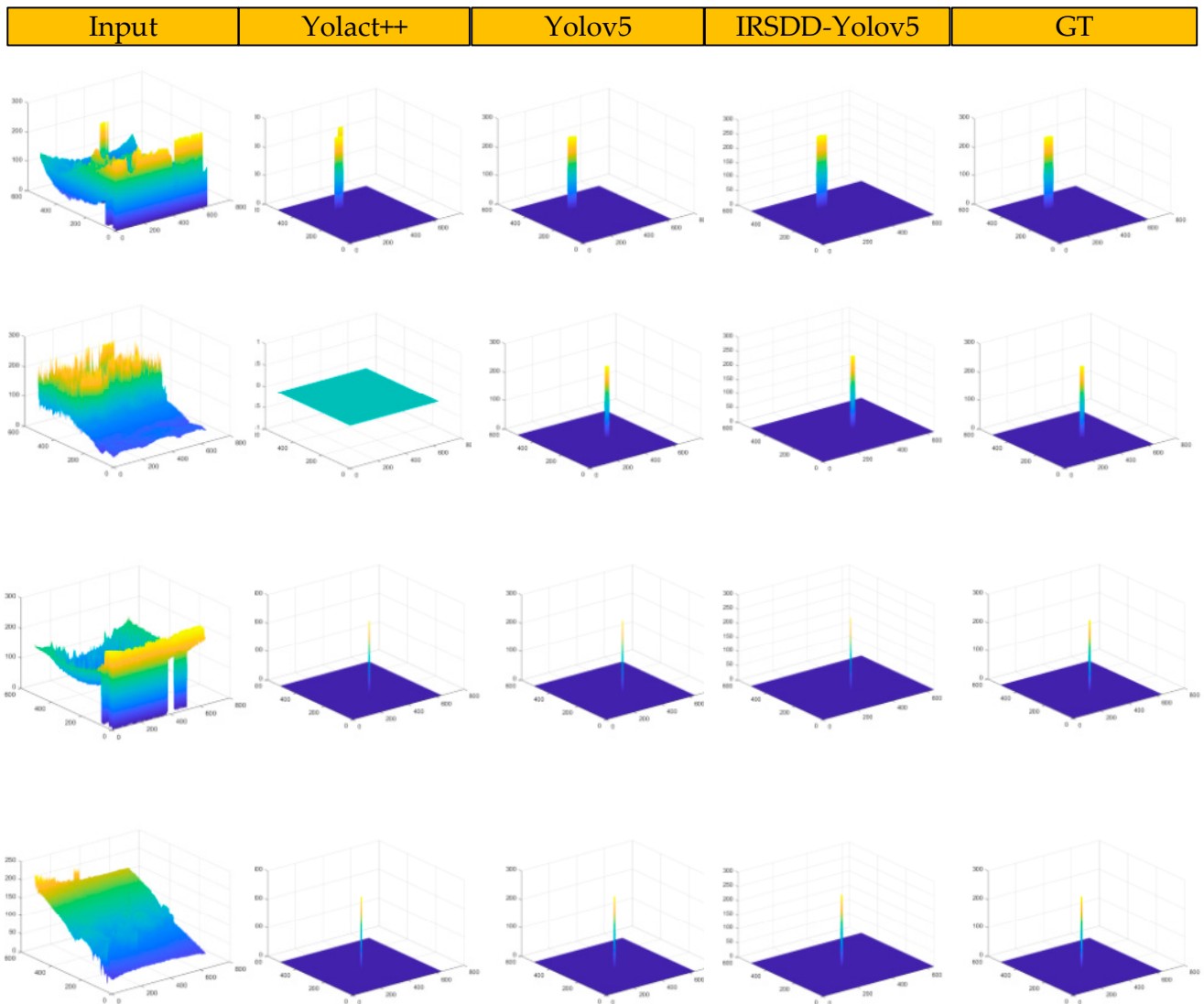

**Figure 10.** Three-dimensional visualization qualitative results of different instance segmentation methods. From left to right is the input original image, the segmentation results of Yolact++, YOLOv5, IRSDD-YOLOV5. The real area of the target (GT). From top to bottom are the city scene, mountain scene, sea scene and sky scene.

### 3.3.3. Analysis of the Reasons for Detection Accuracy

Due to the variability in the detection accuracy in different scenes, we quantitatively and qualitatively analyzed the relationships between the drone targets and backgrounds in different scenes and evaluated the impacts of noise and interference sources on detection accuracy in different scenes.

As can be seen from Table 5, the signal-to-noise ratio (SNR) of the target in the sky scene is the highest, followed by the sea background, while the target SNR in the image with a mountain background is the lowest. As can be seen from Figure 6, in the global three-dimensional gray scale image of the sky scene, there is less background clutter in the whole image, and the distinction between the target and the surrounding background is clear, followed by the city. However, in the mountain and sea scenes, the entire images contain significant amounts of clutter noise. In particular, it can be seen from the three-dimensional local gray scale that the drone target boundary is not clear, and there is a large amount of clutter near the target in the mountain scene.

**Table 5.** Local signal-to-noise ratios of targets and backgrounds for different scenes. Sec1–Sec5 are five representative single-frame images in different scenes, and $\overline{SCR}$ is the average values of local signal-to-noise ratios of five single-frame images.

| Scenes | | SCR | | SCR | | SCR | | SCR | | SCR | $\overline{SCR}$ |
|---|---|---|---|---|---|---|---|---|---|---|---|
| Urban | Sec1 | 2.950 | Sec2 | 1.848 | Sec3 | 6.366 | Sec4 | 0.686 | Sec5 | 0.622 | 2.494 |
| Mountain | Sec1 | 0.880 | Sec2 | 0.506 | Sec3 | 0.283 | Sec4 | 1.280 | Sec5 | 1.822 | 0.954 |
| Sea | Sec1 | 5.219 | Sec2 | 2.255 | Sec3 | 3.952 | Sec4 | 4.852 | Sec5 | 3.744 | 4.004 |
| Sky | Sec1 | 3.769 | Sec2 | 5.898 | Sec3 | 3.790 | Sec4 | 2.600 | Sec5 | 7.594 | 4.730 |

It can also be seen from the combination of Table 5 and Figure 11 that drone detection in the mountain background image is the most complex, followed by the sea scene, and it is difficult to distinguish the target from the background. Although there are some complex objects in cities, such as buildings, the clutter noise in the city scene is relatively small. There are mainly clouds in the sky image, but there is almost no clutter. Although the target occupies fewer pixels in the image, it can still have relatively distinguishable features.

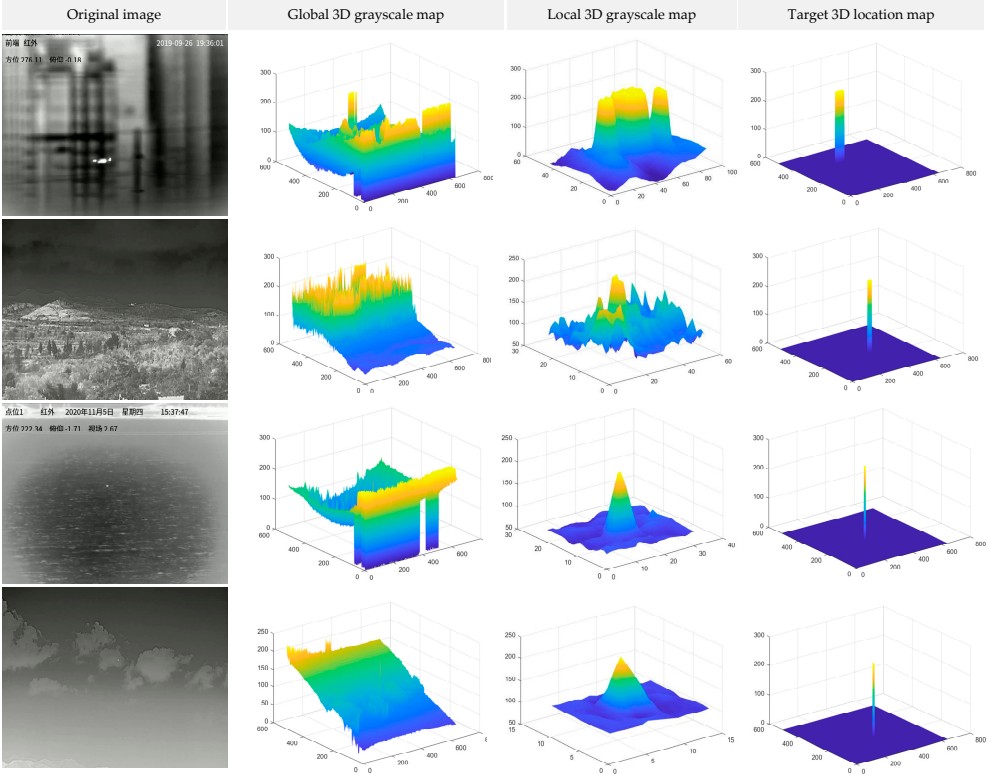

**Figure 11.** Infrared drone images in different scenes compared with different three-dimensional images.

From the above analysis results, we know the reasons for the relatively low detection accuracies in the mountain scene and the sea scene. When generating the training data, although the location and area of the target are manually labeled, there will still be some challenges in detection using the current advanced segmentation methods.

### 3.4. Ablation Studies

In order to demonstrate the effectiveness of the structural design and network module of IRSDD-YOLOv5, different variables were constructed to perform detailed ablation experiments to determine whether the contributions of the different variables were reasonable.

In addition, for the validity and reasonableness of the ablation, we selected mountain and sea scenarios for the ablation experiments. The results of the ablation experiments for the different variables are presented in Tables 6 and 7, where the larger the measured values of $AP_{\{0.5:0.95\}}$, $AP_{50}$ and $AP_s$, the better the performance. The best results in each column of Table 7 are highlighted in bold red font, and the second-best results are highlighted in bold blue font.

**Table 6.** Results of ablation experiments with the infrared small target detection module.

| With/without IRSTDM | Mountain Background | | | Sea Background | | |
|---|---|---|---|---|---|---|
| | $AP_{\{0.5:0.95\}}$ | $AP_{50}$ | $AP_s$ | $AP_{\{0.5:0.95\}}$ | $AP_{50}$ | $AP_s$ |
| With | 0.277 | 0.798 | 0.277 | 0.375 | 0.934 | 0.375 |
| Without | 0.279 | 0.773↓ | 0.279 | 0.339↓ | 0.898↓ | 0.339↓ |

**Table 7.** Results of ablation experiments with loss functions; -- refers to experimental results that do not take into account the NWD loss function.

| β | Mountain Background | | | Sea Background | | |
|---|---|---|---|---|---|---|
| | $AP_{\{0.5:0.95\}}$ | $AP_{50}$ | $AP_s$ | $AP_{\{0.5:0.95\}}$ | $AP_{50}$ | $AP_s$ |
| -- | 0.278 | 0.760 | 0.278 | 0.345 | 0.912 | 0.345 |
| 0.1 | 0.278 | 0.760 | 0.278 | 0.345 | 0.912 | 0.345 |
| 0.2 | 0.275 | 0.767 | 0.275 | 0.346 | 0.914 | 0.346 |
| 0.3 | **0.286** | **0.792** | **0.286** | 0.348 | 0.894 | 0.348 |
| 0.4 | **0.293** | **0.793** | **0.293** | 0.311 | 0.910 | 0.311 |
| 0.5 | 0.284 | 0.783 | 0.284 | 0.351 | **0.911** | 0.351 |
| 0.6 | 0.279 | 0.773 | 0.279 | 0.339 | 0.898 | 0.339 |
| 0.7 | 0.282 | 0.748 | 0.282 | **0.368** | 0.905 | **0.368** |
| 0.8 | 0.286 | 0.771 | 0.286 | **0.359** | **0.934** | **0.359** |
| 0.9 | 0.273 | 0.732 | 0.273 | 0.334 | 0.906 | 0.334 |
| 1.0 | 0.275 | 0.732 | 0.275 | 0.346 | 0.889 | 0.346 |

(1)    Ablation study for small target detection module

We set up an ablation module for the cross-layer feature fusion of infrared small targets to explore the effect of information interaction between shallow detail features and deep semantic features. In the ablation study for the infrared small target detection module (IRSTDM), we fixed the β value of the bounding box loss function to 0.5 by adding or not adding the small target detection module to demonstrate the effectiveness of the small target detection module. The results in the table show that the $AP_{50}$ of IRSDD-YOLOv5 in the mountain scenario decreases by 2.5% if the small target detection module is not added. The AP50 in the mountain scenario decreased by 2.5%, The AP50 in the sea scenario decreased by 3.6%. The experimental results show that the key to improving the network's ability to detect small drones is to maximize the advantages of the small target feature layer.

(2)    Ablation study for loss functions

We considered the advantages and disadvantages of different loss functions and designed the bounding box loss function as shown in Equation (11) by adjusting the value of β in order to explore the impact of optimizing the loss function by combining the CIOU with the NWD. The results in the table show that at a β value of 0.4, the $AP_{\{0.5:0.95\}}$ and $AP_{50}$ measurements of IRSDD-YOLOv5 in the mountain scenario are improved by 1.5% and 3.3%, respectively. The maximum improvement is achieved at a value of β of 0.8, when the AP50 value of IRSDD-YOLOv5 in sea scenes are improved by 2.2%. The experimental

results show that combining the advantages of different loss functions can effectively improve the detection performance of an infrared small target detection network.

### 3.5. Future Outlook

In this section, we discuss in detail the specific trends in infrared drone detection methods based on practical application requirements. Although our proposed IRSDD-YOLOv5 has made significant progress in drone detection based on single-frame infrared images, there are still two issues that need to be considered in future research.

### 3.5.1. Multiple Infrared Drone Detection

In the SIDD dataset for infrared drone detection proposed in this paper, the image size is 640 × 512 pixels, and there is only one target in each image. To evaluate the performance of the proposed IRSDD-YOLOv5 for the infrared detection of multiple drones, we processed the SIDD dataset by stitching four images into one image containing four targets, and the stitching process is shown in Figure 12.

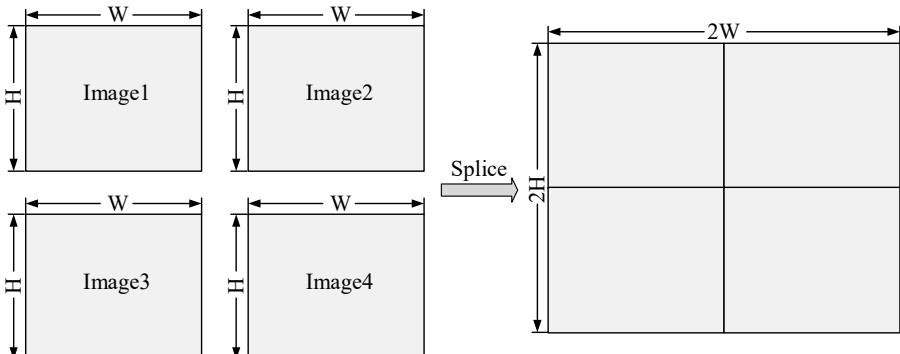

**Figure 12.** The process of stitching four images containing a single object into one image containing four objects.

### 3.5.2. Introduction of a Priori Clue Detection

At present, the identification of drone prevention is usually completed via radar-guided photoelectric or infrared cameras. When the radar detects the drone, the radar cannot accurately obtain the direction of the drone and conduct further tracking. At this time, the radar can provide the infrared camera with the general direction of the drone, and as a prior clue, the algorithm searches and detects small areas in the viewing field. The specific process is shown in Figure 13. Limiting the field of view of an infrared image to the area detected by radar is a common practice in drone detection which helps to improve detection efficiency. Under different experimental conditions, the limit range of an infrared image should be selected according to the actual situation.

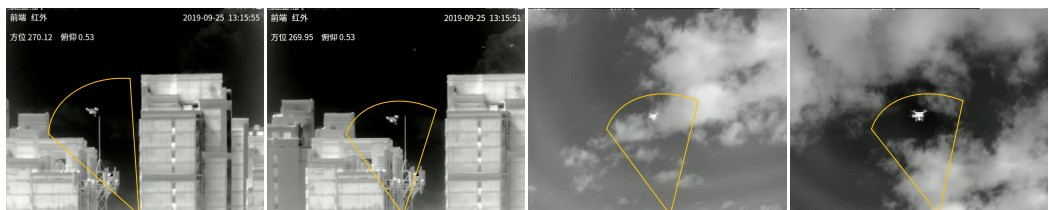

**Figure 13.** An example of narrowing the detection area by introducing prior information from radar. The yellow sector indicates the general location of the drone.

## 4. Conclusions

In this study, we proposed a new infrared small drone detection method, IRSDD-YOLOV5. Firstly, we constructed a small target detection module (IRSTDM) by cascading

the target semantic information of a deep network and the target spatial location information of a shallow network while preserving and focusing the features of an infrared small drone. Secondly, we calculated the Wasserstein distance between two boundary frames based on the Gaussian distribution of the boundary frame so as to optimize the loss function and further improve the accuracy of target detection. Finally, the proposed method and the most advanced methods were trained and tested on the self-made SIDD dataset. The experimental results show that the proposed method achieves excellent performance, among which the $AP_{50}$ measurements of the mountain scene and ocean scene in the dataset reached 79.8% and 93.4%. They were 3.8% and 4% greater than YOLOv5. We also conducted extensive ablation experiments to verify the effectiveness of the proposed method. In addition, we published the SIDD dataset based on instance segmentation detection. However, some unsolved problems, such as multi-target detection and real-time detection, are worth further research.

**Author Contributions:** Conceptualization, S.Y.; methodology, S.Y.; software, S.Y.; validation, H.H. and C.L.; formal analysis, P.W. and S.Y.; investigation, Z.D.; resources, S.Y.; data curation, C.L. and Z.D.; writing—original draft preparation, S.Y.; writing—review and editing, S.Y., H.H. and Z.D.; visualization, S.Y.; supervision, Z.Z. (Zongqing Zhao) and H.H.; project administration, S.Y.; funding acquisition, B.S. and Z.Z. (Zhen Zuo). All authors have read and agreed to the published version of the manuscript.

**Funding:** This research was funded by [National Natural Science Foundation of China] grant number [52101377] And The APC was funded by [National Natural Science Foundation of China].

**Data Availability Statement:** We published the dataset in this paper, which is the first publicly released infrared small UAV dataset with pixel-level labeling. The SIDD dataset is available at https://github.com/Dang-zy/SIDD.git (28 April 2023).

**Conflicts of Interest:** The authors declare no conflict of interest.

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
