# Peer review of "IRSDD-YOLOv5: Focusing on the Infrared Detection of Small Drones"

_drones, doi:10.3390/drones7060393_

Round 1
Reviewer 1 Report
Reviewers Comments
Major revision is being suggested for the manuscript id: drones-2399049, titled, “IRSDD-YOLOv5: Focusing on infrared small drones detection”. The present state of the article appears to exhibit characteristics of a descriptive report with an abundance of descriptive language and a limited amount of research. The usage of numerous adjectives and adverbs may be deemed excessive in relation to the level of analysis provided. The following are specific comments; the author must revise the manuscript and prepare a rebuttal to the comments for further review.
1. The author should refrain from using first-person and third-person pronouns in their research writing.
2. The author must adhere to the provided manuscript template/format of the journal, which can be found at the following link: https://www.mdpi.com/journal/drones/instructions
3. The current abstract fails to provide a clear overview of the presented work. Therefore, the authors are advised to restructure the abstract to include information on problem generalization, research gap, objectives, methodology, major results, and conclusions in a concise and precise manner.
4. Introduction: (i) The first three paragraphs should not contain any citations. (ii) The length of the caption for Figure 1 needs to be checked. (iii) The research gap needs to be identified accurately, and the flow of the manuscript should be reviewed. (iv) The contributions/objectives must be clearly listed in the final paragraph of the introduction, with a maximum of 15-20 words each.
5. The related work can be incorporated into the introduction of the article, which should be rewritten in scientific terminology and made coherent and understandable. The authors may refer to the following articles for better understanding:
(i) https://doi.org/10.3390/drones7050310
(ii) https://doi.org/10.3390/drones7050304
(iii) https://doi.org/10.1002/9781119792109.ch11
(iv) https://doi.org/10.3390/drones7050300
6. Section 2 should be titled Materials and Methods, and the first subsection of the revised section 2 should be Dataset.
7. The proposed method should be presented as a subsection under section 2: Materials and Methods.
8. Section 3 should be titled Results and Discussion, and section 4 should be changed to 3 under the heading.
9. There is no need for sectional descriptions under Results and Discussion. The authors are advised to remove this content.
10. While the authors presented a significant amount of results in tables and figures, and also described them, the research community is interested in obtaining insights from the article. The critical analysis/discussion of the results is lacking, and there is no comparative presentation of the obtained results with published results. The authors must address these shortcomings by conducting a root cause analysis of the obtained results and revising the manuscript accordingly.
11. The caption for Figure 8 needs to be rewritten accurately.
12. Section 4 should be titled Conclusions instead of Conclusion.
13. The authors should rewrite the conclusions in no more than 200 words, focusing only on the important takeaways from the work. Additionally, they may include 1-2 sentences outlining the next directions of research, if desired.
I wish authors a great success.

Author Response
Dear Reviewers,
I fully complied with your suggestions to complete the revision of the paper. The revised version of the paper and the reply to the reviewer are attached.
Thank you for your professional comments and sincere opinions!

Reviewer 2 Report
In this manuscript, the authors propose an IRSDD-YOLOv5 for infrared small drones detection. My specific comments are as follows:
1. Why did the authors not train and validate several scenarios simultaneously?Individual training and validation may not validate the effectiveness of the proposed IRSDD-YOLOv5.
2. Compared to the baseline, the proposed method has a significant improvement in the sea surface scene, but in other cases, there is no significant improvement compared to the baseline.
3. Why is the comparison result for instance segmentation not displayed?
4. Why did Solov2 and BoxInst have no results on Moutain and Sea surface scenes?
5. I think the problem with ablation experiments is significant, and the authors need to explain why the baseline of the two ablations does not match.
6. The batch size used for training in Implementation details is not specified.
7. Missing description of L seg loss in Section 3.3.
8. It is recommended to represent the prediction box and segmented mask header of NN in Figure 2, and provide detailed explanations.
9. The authors could publish the IRSDD-YOLOv5 together with the SIDD dataset.
Author Response
Dear Reviewer,
I fully complied with your suggestions to complete the revision of the paper. The answers and revisions to your comments can be found in the attachment.
Thank you for your professional comments and sincere opinions!

Reviewer 3 Report
The paper proposes a modified YOLOv5 model for focusing on infrared small drones detection, and it is innovative to some extent. But the content needs to be promoted, specifically as follows:
1. The content of section 2 is introductory and should be included in the introduction, and the content of the merged introduction should be reduced. In addition, pay more attention to errors of words and symbols. Such as 2.2(1), line 4 "noise z" has an extra letter "z", line 5 "generator and discriminator" the first letter of "generator" should be capitalized.
2. The formula (1) - (11) and its character symbol annotation "X", "Y", "H", "Z", "B", "X", "O", "S", "B" should be unified use italic.
3. Section 3.1 should explain the overall network architecture in more detail. In addition, which specific model does YOLOv5 use? s, m, or l? What does C3 stand for?
4. Section 3.2 and 3.3 should be combined with formulas to explain the process of pixel processing in more detail, rather than just a large list of formulas.
5. It is suggested to add the example pictures of the dataset in section 4.1, and the introduction of computer hardware in section 4.3. The numbers in Figure 5 are not clear, so it is suggested to enlarge them.
6. In the test evaluation of section 4, it is suggested that the author integrate the content of 4.5 and discuss the test in detail by combining the charts, figures. In addition, there are two 4.5 titles and a number of "Qualitative results" in section 4. The reviewer suggests the author carefully review the manuscript before submission and avoid unnecessary small mistakes.
Average.
Author Response
Dear Reviewer,
I fully complied with your suggestions to complete the revision of the paper. The revised version of the paper and the reply to the reviewer are attached.
Thank you for your professional comments and sincere opinions!

Reviewer 4 Report
The manuscript is well-written and organised. It begins by stating the problem the paper addresses, which is the need for an efficient and effective method for detecting drones in infrared images. The paper describes the proposed solution, a deep-learning model called IRSDD-YOLOv5. The model is divided into two stages: feature extraction and target prediction. In the feature extraction stage, the model analyzes the spatial location and semantic information of the infrared small target in the deep network. In the target prediction stage, the model uses the small target prediction head to predict the prior information output by the infrared small target detection module (IRSTDM). Finally, the Wasserstein Distance between the real frame and the target frame is calculated to optimize the loss function and improve the infrared small target detection performance.
In addition, The authors produced a new dataset (SIDD) and compared the performance of IRSDD-YOLOv5 with YOLOv5.
Finally, the authors compared their proposed method with other well-known methods in the literature, such as Blendmask, BoxInst, Yolact++, Solov2, Mask-Rcnn, Yolov5, Yolov7.
As a result, I think that my work can be accepted as it is.
Author Response
Dear Reviewer,
Thank you for your professional comments and sincere opinions!
Best wishes from authors of drones-2399049!
Round 2
Reviewer 1 Report
Reviewers Comments
The manuscript titled, “IRSDD-YOLOv5: Focusing on infrared small drones detection”, with the Manuscript id: Drones-2399049, is updated as per the provided comments. I recommend that the manuscript be accepted.
I wish the authors great success.

Reviewer 2 Report
The authors have basically solved all my questions